# Mapping global environmental suitability for Zika virus

Jane P Messina[1*], Moritz UG Kraemer[1], Oliver J Brady[2], David M Pigott[2,3], Freya M Shearer[2], Daniel J Weiss[1], Nick Golding[4], Corrine W Ruktanonchai[5], Peter W Gething[1], Emily Cohn[6], John S Brownstein[6], Kamran Khan[7,8], Andrew J Tatem[5,9], Thomas Jaenisch[10,11], Christopher JL Murray[3], Fatima Marinho[12], Thomas W Scott[13], Simon I Hay[2,3*]

[1]Department of Zoology, University of Oxford, Oxford, United Kingdom; [2]Wellcome Trust Centre for Human Genetics, University of Oxford, Oxford, United Kingdom; [3]Institute for Health Metrics and Evaluation, University of Washington, Seattle, United States; [4]Department of BioSciences, University of Melbourne, Parkville, United Kingdom; [5]WorldPop project, Department of Geography and Environment, University of Southampton, Southampton, United Kingdom; [6]Boston Children's Hospital, Harvard Medical School, Boston, United Kingdom; [7]Department of Medicine, Division of Infectious Diseases, University of Toronto, Toronto, Canada; [8]Li Ka Shing Knowledge Institute, St Michael's Hospital, Toronto, Canada; [9]Flowminder Foundation, Stockholm, Sweden; [10]Section Clinical Tropical Medicine, Department for Infectious Diseases, Heidelberg University Hospital, Heidelberg, Germany; [11]German Centre for Infection Research (DZIF), Heidelberg partner site, Heidelberg, Germany; [12]Secretariat of Health Surveillance, Ministry of Health Brazil, Brasilia, Brazil; [13]Department of Entomology and Nematology, University of California Davis, Davis, United States

*For correspondence: jane.
messina@zoo.ox.ac.uk (JPM);
sihay@uw.edu (SIH)

Competing interest: See
page 13

Reviewing editor: Mark Jit,
London School of Hygiene &
Tropical Medicine, and Public
Health England, United Kingdom

**Abstract** Zika virus was discovered in Uganda in 1947 and is transmitted by Aedes mosquitoes, which also act as vectors for dengue and chikungunya viruses throughout much of the tropical world. In 2007, an outbreak in the Federated States of Micronesia sparked public health concern. In 2013, the virus began to spread across other parts of Oceania and in 2015, a large outbreak in Latin America began in Brazil. Possible associations with microcephaly and Guillain-Barré syndrome observed in this outbreak have raised concerns about continued global spread of Zika virus, prompting its declaration as a Public Health Emergency of International Concern by the World Health Organization. We conducted species distribution modelling to map environmental suitability for Zika. We show a large portion of tropical and sub-tropical regions globally have suitable environmental conditions with over 2.17 billion people inhabiting these areas.

## Introduction

Zika virus (ZIKV) is an emerging arbovirus carried by mosquitoes of the genus *Aedes* (*Musso et al., 2014*). Although discovered in Uganda in 1947 (*Dick et al., 1952*; *Dick, 1953*) ZIKV was only known to cause sporadic infections in humans in Africa and Asia until 2007 (*Lanciotti et al., 2008*), when it caused a large outbreak of symptomatic cases on Yap island in the Federated States of Micronesia (FSM), followed by another in French Polynesia in 2013–14 and subsequent spread across Oceania (*Musso et al., 2015a*). In the 2007 Yap island outbreak, it was estimated that approximately 20% of ZIKV cases were symptomatic. While indigenous transmission of ZIKV to humans was reported for

**eLife digest** Zika virus is transmitted between humans by mosquitoes. The majority of infections cause mild flu-like symptoms, but neurological complications in adults and infants have been found in recent outbreaks.

Although it was discovered in Uganda in 1947, Zika only caused sporadic infections in humans until 2007, when it caused a large outbreak in the Federated States of Micronesia. The virus later spread across Oceania, was first reported in Brazil in 2015 and has since rapidly spread across Latin America. This has led many people to question how far it will continue to spread. There was therefore a need to define the areas where the virus could be transmitted, including the human populations that might be risk in these areas.

Messina et al. have now mapped the areas that provide conditions that are highly suitable for the spread of the Zika virus. These areas occur in many tropical and sub-tropical regions around the globe. The largest areas of risk in the Americas lie in Brazil, Colombia and Venezuela. Although Zika has yet to be reported in the USA, a large portion of the southeast region from Texas through to Florida is highly suitable for transmission. Much of sub-Saharan Africa (where several sporadic cases have been reported since the 1950s) also presents an environment that is highly suitable for the Zika virus. While no cases have yet been reported in India, a large portion of the subcontinent is also suitable for Zika transmission.

Over 2 billion people live in Zika-suitable areas globally, and in the Americas alone, over 5.4 million births occurred in 2015 within such areas. It is important, however, to recognize that not all individuals living in suitable areas will necessarily be exposed to Zika.

We still lack a great deal of basic epidemiological information about Zika. More needs to be known about the species of mosquito that spreads the disease and how the Zika virus interacts with related viruses such as dengue. As such information becomes available and clinical cases become routinely diagnosed, the global evidence base will be strengthened, which will improve the accuracy of future maps.

the first time in Latin America in 2015 (*Zanluca et al., 2015*; *World Health Organisation, 2015*), recent phylogeographic research estimates that the virus was introduced into the region between May and December 2013 (*Faria et al., 2016*). This recent rapid spread has led to concern that the virus is following a similar pattern of global expansion to that of dengue and chikungunya (*Musso et al., 2015a*).

ZIKV has been isolated from 19 different *Aedes* species (*Haddow et al., 2012*; *Grard et al., 2014*), but virus has been most frequently found in *Ae. aegypti* (*Monlun et al., 1992*; *Marchette et al., 1969*; *Smithburn, 1954*; *Pond, 1963*; *Faye et al., 2008*; *Foy et al., 2011b*; *WHO Collaborating Center for Reference and Research on Arboviruses and Hemorrhagic Fever Viruses: Annual Report, 1999*). These studies were based upon ancestral African strains of ZIKV, but the current rapid spread of ZIKV in Latin America is indicative of this highly efficient arbovirus vector (*Marcondes and Ximenes, 2015*). The relatively recent global spread of *Ae. albopictus* (*Benedict et al., 2007*; *Kraemer et al., 2015c*) and the rarity of ZIKV isolations from wild mosquitoes may also partially explain the lower frequency of isolations from *Ae. albopictus* populations. Whilst virus transmission by *Ae. albopictus* and other minor vector species has normally resulted in only a small number of cases (*Kutsuna et al., 2015*; *Roiz et al., 2015*), these vectors do pose the threat of limited transmission (*Grard et al., 2014*). The wide geographic distribution of *Ae. albopictus* combined with the frequent virus introduction *via* viraemic travellers (*McCarthy 2016*; *Bogoch et al., 2016*; *Morrison et al., 2008*; *Scott and Takken, 2012*), means the risk for ZIKV infection *via* this vector must therefore also be considered in ZIKV mapping.

The fact that ZIKV reporting was limited to a few small areas in Africa and Asia until 2007 means that global risk mapping has not, until recently, been a priority (*Pigott et al., 2015b*). Recent associations with Guillain-Barré syndrome in adults and microcephaly in infants born to ZIKV-infected mothers (*World Health Organisation, 2015*; *Martines et al., 2016*) have revealed that ZIKV could lead to more severe complications than the mild rash and flu-like symptoms that characterize the

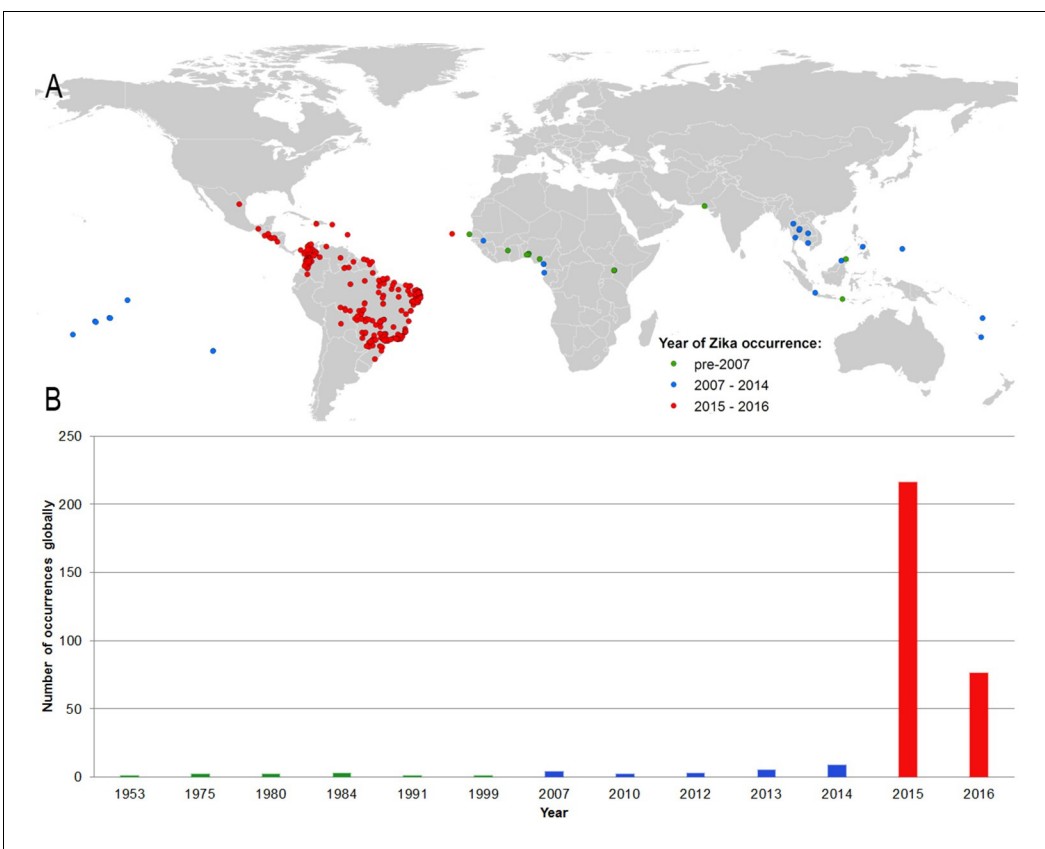

**Figure 1.** (A) Map showing the distribution of the final set of 323 ZIKV occurrence locations entered into the ensemble Boosted Regression Tree modelling procedure. Locations are classified by year of occurrence to show those which took place (i) prior to the 2007 outbreak in Federated States of Micronesia; (ii) between 2007–2014; and (iii) during the 2015–2016 outbreak; (B) the total number of locations reporting symptomatic ZIKV occurrence in humans globally over time.

The following figure supplement is available for figure 1:

**Figure supplement 1.** Maps of all covariates entered into the 300 BRT models.

majority of symptomatic cases (*Gatherer and Kohl, 2016*). Considering these potentially severe complications and the rapid expansion of ZIKV into previously unaffected areas, the global public health community needs information about those areas that are environmentally suitable for transmission of ZIKV to humans. Being a closely related flavivirus to DENV, there is furthermore the potential for antigen-based diagnostic tests to exhibit cross-reactivity when IgM ELISA is used for rapid diagnosis. Although ZIKV-specific serologic assays are being developed by the U.S. Centers for Disease Control, currently the only method of confirming ZIKV infection is by using PCR on acute specimens (Lanciotti et al., 2008, *Faye et al., 2008*). Awareness of suitability for transmission is essential if proper detection methods are to be employed.

In this paper, we use species distribution modelling techniques that have been useful for mapping other vector-borne diseases such as dengue (*Bhatt et al., 2013*), Leishmaniasis (*Pigott et al., 2014b*), and Crimean-Congo Haemorrhagic Fever (*Messina et al., 2015b*) to map environmental suitability for ZIKV. The environmental niche of a disease can be identified according to a combination of environmental conditions supporting its presence in a particular location, with statistical modelling then allowing this niche to be described quantitatively (*Kraemer et al., 2016*). Niche modelling uses records of known disease occurrence alongside hypothesized environmental covariates to predict suitability for disease transmission in regions where it has yet to be reported (*Elith and Leathwick, 2009*). Contemporary high spatial-resolution global data representing a

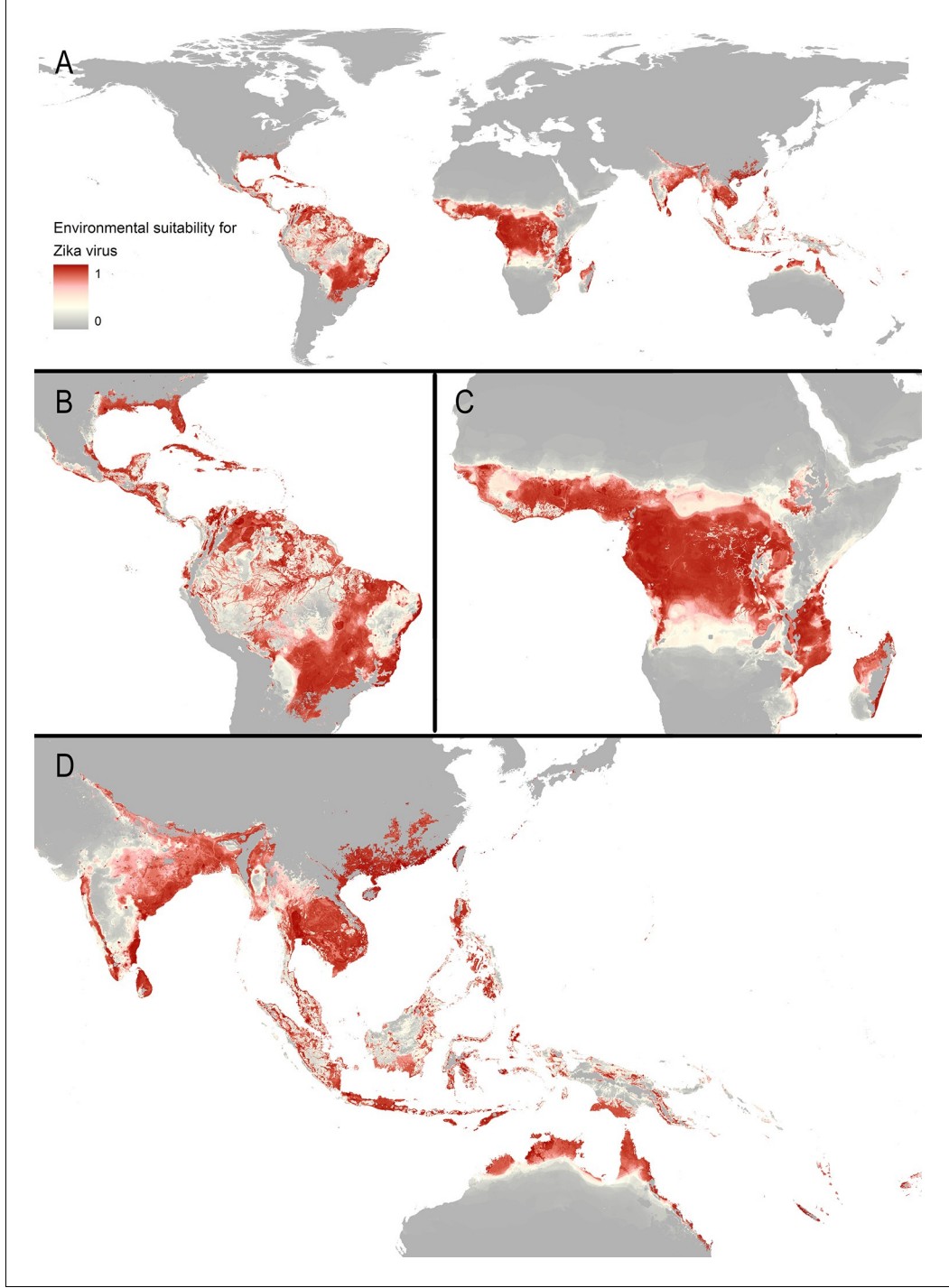

**Figure 2.** Maps of (**A**) global environmental suitability for ZIKV, ranging from 0 (grey) to 1 (red), showing greater detail for (**B**) the Americas, (**C**) Africa, and (**D**) Asia and Oceania.
The following figure supplements are available for figure 2:

**Figure supplement 1.** Uncertainty around Zika suitability predictions displayed in main manuscript – **Figure 2**, ranging from less than 0.01 (very little uncertainty) to 0.94 (greatest uncertainty).
**Figure supplement 2.** Effect plots for each covariate entered into the ensemble of 300 BRT models.

*Figure 2 continued on next page*

*Figure 2 continued*

**Figure supplement 3.** Environmental suitability for Zika virus transmission to humans, not taking into account temperature suitability for dengue *via Aedes albopictus*.
**Figure supplement 4.** Map showing areas predicted to have greater dengue suitability (from *Bhatt et al., 2013*, Nature) vs those which are predicted to have greater Zika suitability in the current study.

variety of environmental conditions allows for these predictions to be made at a global scale (*Hay et al., 2006*).

## Results

*Figure 1A* shows the locations of the 323 standardized occurrence records in the final dataset, classified by the following date ranges: (i) up until 2006 (before the outbreak in FSM); (ii) between 2007 (the year of the FSM outbreak) and 2014; and (iii) since 2015, the first reporting of ZIKV in the Americas. This map is accompanied by the graph in *Figure 1B*, showing the number of reported occurrence locations globally by year. These figures highlight the more sporadic nature of reporting until recent years, with the majority of occurrences in the dataset (63%) coming from the recent 2015–2016 outbreak in Latin America.

The final map that resulted from the mean of 300 ensemble Boosted Regression Tree (BRT) models is shown in *Figure 2A* (with greater detail shown for each region in *Figures 2B–D*). *Figure 2— figure supplement 1* shows the distribution of uncertainty based upon the upper and lower prediction quantiles from the 300 models. We restricted our models to make predictions only within areas where i) mosquito vectors (in this case *Ae. aegypti*) were able to persist and ii) where temperature was sufficient for arboviral replication within the mosquito. The former of these was calculated by taking the *Ae. aegypti* probability of occurrence (*Kraemer et al., 2015c*) value that incorporated 90% of all known occurrences (*Kraemer et al., 2015b*) (giving a threshold value of 0.8 and greater) while the latter was evaluated using a mechanistic mosquito model (*Brady et al., 2013*; *2014*), which identified regions where arboviral transmission could be sustained for at least 355 days (one year

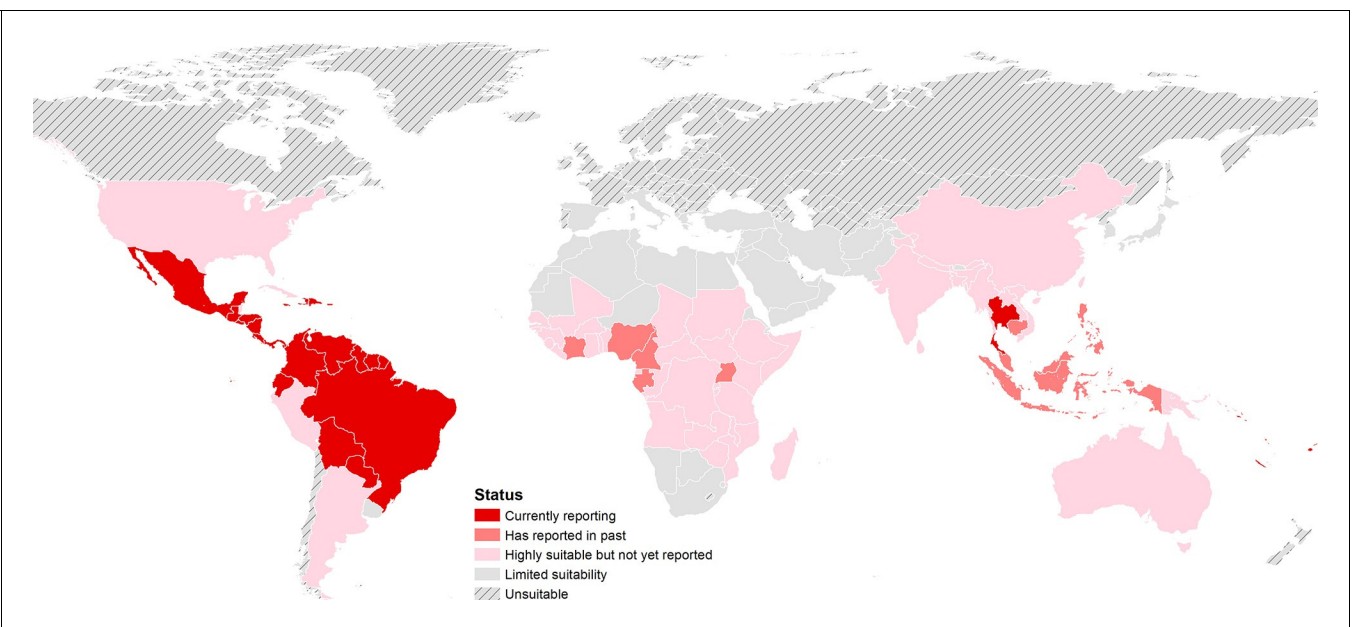

**Figure 3.** Status of ZIKV reporting as of 2016 by country, showing countries that are highly environmentally suitable (having a suitable area of more than 10,000 square kilometres) but which have not yet reported symptomatic cases of ZIKV in humans. 'Currently reporting' countries are those having reported cases since 2015.

minus the human incubation period) in an average year. *Figure 3* is a country-level map distinguishing between those countries that are currently reporting ZIKV, those which have reported ZIKV in the past, those which have highly suitable areas for transmission, and those which are unsuitable. Our models predicted high levels of risk for ZIKV in many areas within the tropical and sub-tropical zones. Large portions of the Americas are suitable for transmission, with the largest areas of risk occurring in Brazil, followed by Colombia and Venezuela, all of which have reported high numbers of cases in the 2015–2016 outbreak. In Brazil, where the highest numbers of ZIKV are reported in the ongoing epidemic, the coastal cities in the south as well as large areas of the north are identified to have the highest environmental suitability of ZIKV. The central region of Brazil, on the other hand, has low population densities and smaller mosquito populations, which is reflected in the relatively low suitability for ZIKV transmission seen in the map. Although ZIKV has yet to be reported in the USA, a large portion of the southeast region of the country, including much of Texas through to Florida, is also highly suitable for transmission. Potential risk for ZIKV transmission is high in much of sub-Saharan Africa, with continuous suitability in the Democratic Republic of Congo and surrounding areas and several sporadic case reports in western sub-Saharan countries since the 1950s. Although no symptomatic cases have yet been reported in India, a large portion of this country is at potential risk for ZIKV transmission (over 2 million square kilometres), with environmental suitability extending from its northwest regions through to Bangladesh and Myanmar. The Indochina region, southeast China, and Indonesia all have large areas of environmental suitability as well, extending into Oceania. While only representing less than ten percent of Australia's total land area, the area shown to be suitable for ZIKV transmission in its northernmost regions is considerable (comprising nearly 250,000 square kilometres).

Our models showed ZIKV risk to be particularly influenced by annual cumulative precipitation, contributing 65.0% to the variation in the ensemble of models. The next most important predictor in the model was temperature suitability for DENV transmission *via Ae. albopictus,* contributing 14.6%. These are followed by urban extents (8.3%), temperature suitability for DENV *via Ae. aegypti* (5.7%), the Enhanced Vegetation Index (EVI; 3.8%), and minimum relative humidity (2.5%). Effect plots for each covariate are provided in *Figure 2—figure supplement 2*. Validation statistics indicated high predictive performance of the BRT ensemble mean map evaluated in a 10-fold cross-validation procedure, with area under the receiver operating characteristic (AUC) of 0.829 ( ± 0.121 SD). Due to

**Table 1.** Population living in areas suitable for ZIKV transmission within each major world region and top four countries contributing to these populations at risk.

| Region/Country | Population living in areas suitable for ZIKV transmission (millions) |
| --- | --- |
| Africa | 452.58 |
| Nigeria | 111.97 |
| Democratic Republic of the Congo | 68.95 |
| Uganda | 33.43 |
| United Republic of Tanzania | 22.70 |
| Americas | 298.36 |
| Brazil | 120.65 |
| Mexico | 32.22 |
| Colombia | 29.54 |
| Venezuela | 22.22 |
| Asia | 1,422.13 |
| India | 413.19 |
| Indonesia | 226.04 |
| China | 213.84 |
| Bangladesh | 133.29 |
| World | 2,173.27 |

the uncertainty about *Ae. albopictus* as a competent vector for ZIKV, we also provide results for an ensemble of models which did not include temperature suitability for dengue *via* this mosquito species in *Figure 2—figure supplement 3*.

A threshold environmental suitability value of 0.397 in our final map was determined to incorporate 90% of all ZIKV occurrence locations. This was used to classify each 5 km x 5 km pixel on our final map as suitable or unsuitable for ZIKV transmission to humans. Using high-resolution global population estimates (*WorldPop, 2015*; *SEDAC, 2015*), we summed the populations living in Zika-suitable areas and have identified 2.17 billion people globally living within areas that are environmentally suitable for ZIKV transmission. *Table 1* shows a breakdown of this figure by major world region, also showing the top four contributing countries to the potential population at risk. Asia has the most people living in areas that are suitable for ZIKV transmission at 1.42 billion, accounted for in large part by those living in India. In Africa, roughly 453 million people are living in areas suitable for ZIKV transmission, the largest proportion of which live in Nigeria. In the Americas, more than 298 million people live in ZIKV-suitable transmission zones, with approximately 40 percent of these people living in Brazil. Within the majority of environmentally suitable areas for ZIKV in the Americas, prolonged year-round transmission is possible. Southern Brazil and Argentina, however, are more likely to see transmission interrupted throughout the year, as is the case with the USA should autochthonous ZIKV transmission occur there. Using high-resolution data on births for the year 2015 (*WorldPop, 2015*), we also estimate that 5.42 million births will occur in the Americas over the next year within areas and times of environmental suitability for ZIKV transmission.

## Discussion

A large number of viruses (circa 219) are known to be pathogenic (*Woolhouse et al., 2012*). Of the 53 species of *Flavivirus*, 19 are reported to have caused illness in humans (*ICTV, 2014*). Some flaviviruses, such as DENV, YFV, Japanese encephalitis virus, and West Nile virus, are widespread, causing many thousands of infections each year. The remainder, however, have been recognized as being pathogenic to humans for decades, but have highly focal reported distributions and are only minor contributors to mortality and disability globally (*Hay et al., 2013*; *Murray et al., 2015*). As a result, many are of relatively low priority when research and policy interest are considered (*Pigott et al., 2015b*). The recent spread of ZIKV across the globe highlights the need to reassess our consideration of these other flaviviruses, to gain a better understanding of the factors driving their spread and the potential for geographic expansion beyond their currently limited geographical extents.

Environmental suitability for virus transmission in an area does not necessarily mean that it will arrive and/or establish in that location. Arboviral infections in particular are dependent on a variety of non-environmental factors, with their movement having historically been largely attributed to human mobility from travel, trade, and migration, which introduce the viruses to places where mosquito vectors are already present (*Murray et al., 2013*; *Weaver and Reisen, 2010*; *Nunes et al., 2015*; *Gubler and Clark, 1995*). The identification of locations with permissible environments for transmission of emerging diseases like ZIKV is crucial, as importation could give rise to subsequent autochthonous cases in these locations (*Hennessey et al., 2016*; *Zanluca et al., 2015*). In order to identify places potentially receptive for ZIKV, we assembled the first comprehensive spatial dataset for ZIKV occurrence in humans and compiled a comprehensive set of high-resolution environmental covariates. We then used these data to implement a species distribution modelling approach (*Elith and Leathwick, 2009*) that has proven useful for mapping other vector-borne diseases (*Bhatt et al., 2013*; *Pigott et al., 2014a*; *Mylne et al., 2015*; *Messina et al., 2015b*), allowing us to make inferences about environmental suitability for ZIKV transmission in areas where it has yet to be reported or where we are less certain about its presence. How the ongoing epidemic unfolds in terms of case numbers (or incidence) will depend on a range of other factors such as local transmission dynamics, herd immunity, patterns of contact among mosquitoes and infectious and susceptible humans (*Stoddard et al., 2013*), and mosquito-to-human ratios as recently shown for dengue (*Kraemer et al., 2015a*) and chikungunya (*Salje et al., 2016*).

Globally, we predict that over 2.17 billion people live in areas that are environmentally suitable for ZIKV transmission. We also estimate the number of births occurring in the Americas only, as it is the region for which the most accurate high-resolution population data on births exists (*Tatem et al., 2014*; *Sorichetta et al., 2015*) and because it is the focus of an ongoing outbreak,

which is the largest recorded thus far. In the Americas alone, an estimated 5.42 million births occurred in 2015 within areas and at times that are suitable for ZIKV transmission. It is important to recognize that not all individuals will be exposed to ZIKV. Like with other flaviviruses, a ZIKV outbreak may be temporally and spatially sporadic and, even in the most receptive environments, is unlikely that all of the population will be infected. Furthermore, increasing herd immunity of this likely sterilizing infection will rapidly reduce the size of the susceptible population at risk for infection in subsequent years (*Dick et al., 1952*) and work is ongoing to predict the likely infection dynamics after establishment. Instead, the estimates are intended as indicators of the total number of individuals or births that may require protection during the first wave of the outbreak. Specifically, these populations should be the focus of efforts to increase awareness and provide guidelines for mitigating personal risk of infection. In future analyses, our estimates could be extended to include ZIKV incidence and the virus' effect on incidence of associated conditions such as Guillain-Barré syndrome and microcephaly. Before appropriately caveated estimates can be generated, however, more information is needed regarding: (i) the background rate of these conditions due to other causes; (ii) how risk may vary throughout the course of a pregnancy; (iii) the proportion of the population exposed during outbreaks; and (iv) whether or not immunity acquired through a mother's prior exposure is protective.

For all arboviral diseases, public health education about reducing populations and avoiding contact with mosquito vectors is required in at-risk areas. Specific to ZIKV is the risk of microcephaly in newborns, which has led public health agencies to issue warnings for women who are currently or planning on becoming pregnant in areas suspected to have ongoing ZIKV transmission and the declaration of a Public Health Emergency of International Concern (*Heymann et al., 2016*). Due to the sensitive nature and implications of these warnings, it is important that levels of risk are rigorously estimated, validated, and updated. Transmission of related arboviral diseases still occurs in many areas we defined as at-risk for ZIKV, which highlights the need for improved vector control outcomes, particularly those targeting *Ae. aegypti*. Predicted levels of risk for ZIKV transmission are potentially helpful for prioritized allocation of vector control resources, as well as for differential diagnosis and, if a vaccine becomes available, delivery efforts. It should be noted that instances of ZIKV sexual transmission have been reported (*Patino-Barbosa et al., 2015*; *Musso et al., 2015b*; *Foy et al., 2011a*). We did not incorporate secondary modes of transmission into the models we described here, but our map can help inform future discussions about the potential impact of this mode of transmission as its relative importance becomes better understood.

A great deal of basic epidemiological information specific to ZIKV is lacking. As a result, information must be leveraged from our knowledge about transmission of related arboviruses. Previous work has focused on mapping other vector borne diseases that share much of the ecology of Zika, such as DENV (*Bhatt et al., 2013*) and CHIKV, as well as for its primary vectors, *Ae. aegypti* and *Ae. albopictus* (*Kraemer et al., 2015c*). For this reason, temperature suitability for dengue (*Brady et al., 2013*, *2014*) was entered into the models due to the greater number of field and laboratory studies available for parameterising this metric for DENV. Until more studies related to vector competence and temperature constraints on ZIKV transmission to humans are conducted, this is the most accurate indicator of arboviral disease transmission *via Aedes* mosquitoes currently available. Indeed, all other covariates in our models could equally be applied to mapping DENV and CHIKV, and ZIKV-specific refinements to modelling covariates will be possible as the disease continues to expand to allow for improvements in future iterations of the map. The relatively smaller amount of occurrence data available for ZIKV (especially prior to recent outbreaks) means that this dataset should also be updated with new information as necessary, leading to a stronger global evidence base and improved accuracy of future maps. Better understanding of ZIKV transmission dynamics will eventually allow for further cartographic refinements to be made, such as the differentiation between endemic- and epidemic-prone areas. Still, all covariates included in the current study have been updated and refined since (*Bhatt et al., 2013*), and when combined with the most extensive occurrence database available for ZIKV, the resulting map we present here is currently the most accurate depiction of the distribution of environmental suitability for ZIKV. A map highlighting differences in predicted suitability for both diseases is provided in *Figure 2—figure supplement 4*.

## Conclusion

In this study, we produced the first global high spatial-resolution map of environmental suitability for ZIKV transmission to humans using an assembly of known records of ZIKV occurrence and environmental covariates in a species distribution modelling framework. While it is clear that much remains to be understood about ZIKV, this first map serves as a baseline for understanding the change in the geographical distribution of this globally emerging arboviral disease. Knowledge of the potential distribution can encourage more vigilant surveillance in both humans and *Aedes* mosquito populations, as well as help in the allocation of limited resources for disease prevention. Public health awareness campaigns and advice for mitigation of individual risk can also be focused in the areas we have predicted to be highly suitable for ZIKV transmission, particularly during the first wave of infection in a population. The maps we have presented may also inform existing travel advisories for pregnant women and other travellers. The maps and underlying data are freely available online *via* figshare (http://www.figshare.com).

# Materials and methods

To map environmental suitability for ZIKV transmission to humans, we applied a species distribution modelling approach to establish a multivariate empirical relationship between the probability of ZIKV occurrence and the environmental conditions in locations where the disease has been confirmed. We employed an ensemble boosted regression trees (BRT) methodology (*De'ath, 2007*; *Elith et al., 2008*), which required the generation of: (i) a comprehensive compendium of known locations of disease occurrence in humans; (ii) a set of background points representing locations where ZIKV has not yet been reported; and (iii) a set of high-resolution globally gridded environmental and socioeconomic covariates hypothesised to affect ZIKV transmission. The resulting model produces a 5 x 5 km spatial-resolution global map of environmental suitability for ZIKV transmission to humans.

## Assembly of the geo-referenced ZIKV occurrence dataset

Information about the locations of ZIKV occurrence in humans was extracted from peer-reviewed literature, case reports, and informal online sources following previously established protocols (*Kraemer et al., 2015b*; *Messina et al., 2014*; *2015a*). To collate the peer-reviewed dataset, literature searches were undertaken using PubMed (http://www.ncbi.nlm.nih.gov/pubmed) and ISI Web of Science (http://www.webofknowledge.com) search engines using the search term 'Zika'. No language restrictions were placed on these searches; however, only those citations with a full title and abstract were retrieved, resulting in the review of 148 references ranging in publication dates between 1951 and 2015. In-house language skills allowed review of all English, French, Portuguese and Spanish articles for useable location information for human ZIKV occurrence. ProMED-mail (http://www.promedmail.org) was also searched using the term 'Zika', resulting in the review of 139 reports between 27 June 2007 and 18 January 2016. Additionally, the most current database of ZIKV case locations in Brazil was obtained directly from the Brazilian Ministry of Health. From all sources, only laboratory confirmation of symptomatic ZIKV infection in humans was entered into the dataset (mention of suspected cases was not entered). Serological evidence from healthy individuals could represent a past infection, with transmission potentially occurring in a different location to that where the individual currently resides (*Darwish et al., 1983*), or could be an artefact from possible cross-reactivity with a variety of different viruses (*Smithburn et al., 1954*). As a result, these less reliable diagnoses of ZIKV were excluded.

All available location information was extracted from each peer-reviewed article and ProMED case report. The site name was used together with all contextual information provided about the site to determine its latitudinal and longitudinal coordinates using Google Maps (https://www.maps.google.com). If the study site could be geo-positioned to a specific place, it was recorded as a point location. If the study site could only be identified at an administrative area level (e.g. province or district), it was recorded as a polygon along with an identifier of its administrative unit. If imported cases were reported with information on the site of infection, they were geo-positioned to this site; if imported cases were reported with no information about the site of infection, they were not entered into the dataset. Informal online data sources were collated automatically by the web-based

system HealthMap (http://www.healthmap.org) as described elsewhere (*Freifeld et al., 2008*). Alerts for ZIKV were obtained from HealthMap for the years 2014–2016, and then manually checked for validity. In total, usable location information was extracted from 110 sources. Information was also collected about the status of symptoms in each reported occurrence, distinguishing between those where symptomatic cases were being reported, versus those where only seroprevalence was detected in healthy individuals.

Due to the potential for multiple independent reports referring to the same cases temporal and spatial standardization was required, as we have described previously in detail for dengue mapping efforts (*Messina et al., 2014*). In brief, an occurrence was defined as a unique location with one or more confirmed cases of ZIKV occurring within one calendar year (the finest temporal resolution available across all records). Point locations were considered to be overlapping if they lay on the same 5 km x 5 km pixel, and polygon locations were identified by a unique administrative unit code. Furthermore, all polygons whose geographic area was greater than one square decimal degree (approximately 111 square kilometers at the equator) were removed from the dataset to avoid averaging covariate values over very large areas, and only those occurrences comprising symptomatic individuals were retained for modelling purposes to ensure an accurate location of infection. In total, the final occurrence dataset contained 323 unique occurrences to be entered into our BRT modelling procedure. A map of the final set of occurrence locations is provided as *Figure 1A*.

## Generation of the background location dataset

Separate maps of the relative probability of occurrence of *Ae. aegypti* and *Ae. albopictus* (*Kraemer et al., 2015c*) were used to compute a combined metric of the relative probability of vector occurrence, by taking the maximum value from the two layers for all 5 km x 5 km gridded cells globally. The inverse of this combined-*Aedes* occurrence probability layer (higher values indicating greater certainty of absence) was then used to draw a biased sample of 10,000 background locations. As such, a greater number of background points were sampled in areas where we are more certain that *Ae. aegypti* or *Ae. albopictus* do not occur, and therefore where ZIKV is less likely to be transmitted to humans. While it has been demonstrated that predictive accuracy from presence-background species distribution models can be improved by biasing background record locations toward areas with greatest reporting probabilities (*Phillips et al., 2009*), information on possible reporting biases, or proxies of such spatial bias, are currently unavailable for ZIKV. These 10,000 background locations were combined with the standardized occurrence dataset to serve as comparison data locations in the BRT species distribution modelling procedure. The background locations were weighted such that their total sum was equal to the total number of occurrence locations (n=237; pseudo-absence weighting=0.0237), in order to aid in the discrimination capacity of the model (*Barbet-Massin et al., 2012*).

## Explanatory covariates

A set of six covariates hypothesized to influence the global distribution of ZIKV transmission to humans were used in our models to establish an empirical relationship between ZIKV presence or absence and underlying environmental conditions. These six covariates included: (i) an index of temperature suitability for dengue transmission to humans *via Ae. aegypti;* (ii) temperature suitability for dengue transmission to humans *via Ae. albopictus*; (iii) minimum relative humidity; (iv) annual cumulative precipitation; (v) an enhanced vegetation index (EVI); and (vi) urban versus rural habitat type. The underlying hypothesis behind each of the covariates is discussed in more detail below, along with a description of data sources and any processing that was undertaken before entering these covariates into our models. Maps of each covariate layer are provided in the supplementary information in *Figure 1—figure supplement 1*.

### Temperature suitability for dengue transmission to humans

via Ae. aegypti *or* Ae. albopictus: Temperature affects key physiological processes in *Aedes* mosquitoes, including age- and temperature-dependent adult female survival, as well as the duration of the extrinsic incubation period (EIP) of arboviruses and the length of the gonotrophic cycle (*Brady et al., 2013*). While these parameters have yet to be measured experimentally for ZIKV, they have been for the closely related DENV. We obtained temperature data from WorldClim v1.03 (http://www.

wordclim.org), which uses historic global meteorological station data from 1961–2005 to interpolate global climate surfaces. MARKSIM software (*Jones and Thornton, 2000*) was then used to apply the coefficients of 17 Global Climate Models (GCMs) to estimate temperature values for the year 2015. This enabled us to incorporate the quantified effects of temperature on DENV transmission into a cohort simulation model that analysed the cumulative effects of both diurnal and inter-seasonal changes in temperature on DENV transmission within an average year, both for *Ae. aegypti* and *Ae. albopictus* separately. The models were then applied to the 2015 temperature data for each 5 km x 5 km grid cell globally. This resulted in maps of temperature suitability for DENV transmission by either *Aedes* species ranging from 0 (no suitable days) to 1 (365 suitable days). These measures were then used as a proxy for temperature suitability for ZIKV transmission to humans.

## Annual cumulative precipitation

Presence of static surface water in natural or man-made containers is a pre-requisite for *Aedes* oviposition and larval and pupal development. While fine-scale spatial and temporal heterogeneities have been observed between precipitation, vector abundance, and incidence of human DENV infections, there is evidence that areas with greater amounts of precipitation are generally associated with higher DENV infection risk (*Chandy et al., 2013*; *Chowell and Sanchez, 2006*; *Dom et al., 2013*; *Pinto et al., 2011*; *Restrepo et al., 2014*; *Sang et al., 2014*; *Sankari et al., 2012*; *Campbell et al., 2015*). Although studies that directly connect levels of precipitation to ZIKV transmission have yet to exist, we assumed for Zika a similar association of precipitation as closely related flaviviruses. WorldClim v1.03 precipitation data and MARKSIM software were used as described above for temperature, to estimate annual cumulative precipitation for the year 2015 for each 5 km x 5 km grid cell globally.

## Minimum relative humidity

Greater relative humidity has been found to promote DENV propagation in *Ae. aegypti* mosquitoes in several localized settings (*Colón-González et al., 2011*; *Thu et al., 1998*), and has also been found to be an important contributor when predicting DENV risk at a global scale (*Hales et al., 2002*). Therefore, we again assumed a similar association for ZIKV in the absence of any direct studies, and included the minimum annual relative humidity in our models as a potential limiting factor to ZIKV transmission. Relative humidity (RH) was calculated as a percent of saturation humidity, or the amount of water vapour required to saturate the air given a particular temperature, using the temperature data from WorldClim v1.03 described earlier. The saturation, or 'dew', point ($T_{dew}$) was calculated using a tabular relationship (*Linacre, 1977*). RH was then calculated as follows:

$$RH = \frac{V(T_x)}{V(T_{dew})} \times 100$$

Where $V(T_{dew}) = 611.21 \times \exp\left(17.502 \times \frac{T}{240.97+T}\right)$ and $V(T_x)$ is the humidity at the given temperature. We then extracted the minimum annual RH for each 5 km x 5 km pixel globally for the year 2015.

## Enhanced vegetation index (EVI)

A close association has been shown between local moisture supply, vegetation canopy development, and abundance of mosquito reproduction (*Linthicum et al., 1999*), with previous studies highlighting the importance of moisture-related measures such as relative humidity to DENV occurrence (*Hales et al., 2002*). Although resistant to desiccation, both *Aedes* eggs and adults require moisture to survive (*Cox et al., 2007*; *Sota and Mogi, 1992*; *Reiskind and Lounibos, 2009*; *Costa et al., 2010*; *Luz et al., 2008*), with low dry season moisture levels substantially affecting *Aedes* mortality (*Russell et al., 2001*; *Trpis, 1972*; *Luz et al., 2008*). Vegetation canopy cover has previously been associated with higher *Aedes* larvae density (*Fuller et al., 2009*; *Troyo et al., 2009*; *Bisset Lazcano et al., 2006*; *Barrera et al., 2006*) by reducing evaporation from containers, decreasing sub-canopy wind speed, and protecting outdoor habitats from direct sunlight. To account for these factors, we included a 5 km x 5 km resolution measure of the EVI derived from NASA's Moderate Resolution Imaging Spectrometer (MODIS, Boulder, Colorado) imagery (*Wan et al., 2002*; *Lin, 2012*), summarized from gap-filled, 8-day, 1 km x 1 km resolution images

acquired globally for years 2000 through 2014 (*Weiss et al., 2014*) to produce a mean annual EVI layer. This mean EVI product is indicative of amount of photosynthesis taking place in the environment over the course of a year, which is positively correlated with the density of vegetation, and is thus a proxy for the level of moisture available given the relationship between precipitation and vegetative growth.

## Urban versus rural habitat type

There is a well-established link between urban areas, some vector borne diseases, and their vectors. In particular, *Ae. aegypti* is found in close proximity to human dwellings often breeding in artificial containers (*Brown et al., 2011*; *Powell and Tabachnick, 2013*; *Kraemer et al., 2015c*). To identify the relationship between urbanisation and ZIKV presence we adapted probabilistic spatial modelling techniques to predict the spatial distribution of global urban extents at a 5 km x 5 km spatial resolution. We used urban growth rates from the United Nations Population Division (*Division, 2014*), paired with urban extents measured and tested by the Moderate Resolution Imaging Spectroradiometer Collection 5 (MODIS C5) land-cover product for Asia (*Schneider et al., 2009*; *2010*; *2015*). A set of spatial covariate datasets hypothesized to influence the spatial patterns of urban expansion was generated, including the time to travel from each 5 km x 5 km pixel to a major city (*Nelson, 2008*), the proportion of urbanised land within a buffer of 20 km, human population density (*Linard and Tatem, 2012*; *Stevens et al., 2015*; *Gaughan et al., 2013*), slope (*Becker et al., 2009*), and distance to water (*Arino et al., 2008*). A BRT modelling approach was then used to predict areas that would become urban in 2015 (*Linard et al., 2013*). Outputs were tested against a training dataset comprising points from Asia only, and showed good overall predictive performance (AUC=0.82). The output raster is a 5 km x 5 km gridded surface with urban (1) vs. rural (0) pixels.

## Ensemble boosted regression trees approach

The boosted regression tree (BRT) modelling procedure combines regression trees with gradient boosting (*Friedman, 2001*). In this procedure, an initial regression tree is fitted and iteratively improved upon in a forward stagewise manner (boosting) by minimising the variation in the response not explained by the model at each iteration. It has been shown to fit complicated response functions efficiently, while guarding against over-fitting by use of extensive internal cross-validation. As such, this approach has been successfully employed in the past to map dengue and its *Aedes* mosquito vectors, as well as other vector-borne diseases (*Bhatt et al., 2013*; *Pigott et al., 2014b*; *Messina et al., 2015b*; *Kraemer et al., 2015c*). To increase the robustness of model predictions and quantify model uncertainty, we fitted an ensemble (*Araújo and New, 2007*) of 300 BRT models to separate bootstraps of the data. We then evaluated the central tendency as the mean across all 300 BRT models (*Bhatt et al., 2013*). Each of the 300 individual models was fitted using the gbm. step subroutine in the dismo package in the R statistical programming environment (*Elith et al., 2008*). All other tuning parameters of the algorithm were held at their default values (tree complexity= 4, learning rate= 0.005, bag fraction= 0.75, step size= 10, cross-validation folds=10). Each of the 300 models predicts environmental suitability on a continuous scale from 0 to 1, with a final prediction map then being generated by calculating the mean prediction across all models for each 5 km x 5 km pixel. Cross-validation was applied to each model, whereby ten subsets of the data comprising 10% of the presence and background observations were assessed based on their ability to predict the distribution of the other 90% of records using the mean area under the curve (AUC) statistic. This AUC value was then averaged across the ten sub-models and finally across all 300 models in the ensemble in order to derive an overall estimate of goodness-of-fit. Additionally, to avoid AUC inflation due to spatial sorting bias, a pairwise distance sampling procedure was used, resulting in a final AUC which is lower than would be returned by standard procedures but which gives a more realistic quantification of the model's ability to extrapolate predictions to new regions (*Wenger and Olden, 2012*). We restricted our models to make predictions only within areas where either *Ae. aegypti* probability of occurrence (*Kraemer et al., 2015c*) is more than 0.8 or temperature is conducive to transmission for at least 355 days in an average year. A second ensemble of 300 models was executed which did not take into account temperature suitability for dengue transmission via *Ae. albopictus*, due to the uncertainty of this species as a competent ZIKV vector. The results of this ensemble of models are provided in *Figure 2—figure supplement 3*.

## Population and births at risk

To calculate the number of people located in an area that is at any level of risk for ZIKV transmission, the global ZIKV environmental suitability map was combined with fine-scale global population surfaces (*SEDAC, 2015*; *WorldPop, 2015*). Firstly, the continuous ZIKV environmental suitability map (ranging from 0 to 1) was converted into a binary surface indicating whether there is any risk of transmission. To do this, we carried out a protocol previously used in (*Pigott et al., 2015a*), choosing a threshold environmental suitability value that encompasses 90% of the ZIKV occurrence point locations. This threshold cut-off of 90% was chosen (rather than 100%) to reflect potential errors or inaccurate locations in the occurrence point dataset. Every 5 km x 5 km pixel in the suitability map with a value above this threshold value was considered at risk for ZIKV transmission. Finally, to estimate the population at risk, we multiplied this binary ZIKV risk map by the global population counts (aligned and aggregated to the same 5 x 5 km grid) for the year 2015 and summed across all cells.

We next estimated the maximum number of births potentially affected by ZIKV in Latin America, as this region is the focus of the recent outbreak and the first to point to a possible association with microcephaly in newborn infants to mothers infected with ZIKV. In order to do this, we first identified the proportion of the year that is suitable for ZIKV transmission within areas that are predicted to be suitable in the binary ZIKV risk map. This proportion was derived from existing temperature suitability models (*Brady et al., 2013*; *2014*), which predict the total number of days within an average year that arbovirus transmission can be sustained in *Ae. aegypti,* assuming there is a local human reservoir of infection. While the intra-mosquito viral dynamics in this model were parameterised for dengue virus, the limited information currently available on other arboviruses suggests that their dynamics are similar (*Lambrechts et al., 2011*). Using the resulting 5 km x 5 km map showing the proportion of the year suitable for ZIKV transmission to humans, we then multiplied this by a map (also at a 5 km x 5 km resolution) of the number of births in the Americas for the year 2015, updated from (*Tatem et al., 2014*; *UNFPA, 2014*). The resulting map indicates the number of births in the Americas potentially at risk for ZIKV (for 2015), assuming ZIKV currently fully occupies its environmental niche and that births are evenly distributed throughout the year.

## Acknowledgements

We thank the Secretariat of Health Surveillance, Ministry of Health of Brazil for providing access to the geographical coordinates of occurrence. JPM, MUGK and TJ receive, and OJB and SIH acknowledge funding from the International research Consortium on Dengue Risk Assessment Management and Surveillance (IDAMS; European Commission 7[th] Framework Programme (21893)). OJB and SIH are supported by the Bill & Melinda Gates Foundation (OPP1053338). SIH is also funded by a Senior Research Fellowship from the Wellcome Trust (095066), and grants from the Bill & Melinda Gates Foundation (OPP1119467, OPP1106023 and OPP1093011). DMP is also funded by the Bill & Melinda Gates Foundation (OPP1093011). DJW and PWG receive support from the Bill and Melinda Gates Foundation (OPP1068048, OPP1106023). NG is supported by a University of Melbourne McKenzie fellowship. CWR is funded through the University of Southampton's Economic and Social Research Council's Doctoral Training Centre. TWS is supported by grants from the National Institutes of Health (P01AI098670) and the Bill and Melinda Gates Foundation (OPP1081737). EC and JSB are supported by the National Library of Medicine of the National Institutes of Health (R01LM010812).

## Additional information

### Competing interests

SIH: Reviewing editor, *eLife*. The other authors declare that no competing interests exist.

### Funding

| Funder | Grant reference number | Author |
|---|---|---|
| European Commission | 21893 | Jane P Messina<br>Moritz UG Kraemer<br>Thomas Jaenisch |

| Bill and Melinda Gates Foundation | OPP1053338; OPP1119467; OPP1106023; OPP1093011; OPP1081737; OPP1068048 | Oliver J Brady David M Pigott Daniel J Weiss Thomas W Scott Simon I Hay |
|---|---|---|
| Wellcome Trust | 095066 | Simon I Hay |
| University of Southampton | Economic and Social Research Council's Doctoral Training Centre | Corrine W Ruktanonchai |
| National Institutes of Health | P01AI098670 | Thomas W Scott |
| University of Melbourne | McKenzie fellowship | Nick Golding |

The funders had no role in study design, data collection and interpretation, or the decision to submit the work for publication.

### Author contributions

JPM, DMP, SIH, Conception and design, Acquisition of data, Analysis and interpretation of data, Drafting or revising the article; MUGK, OJB, NG, Conception and design, Analysis and interpretation of data, Drafting or revising the article; FMS, PWG, Conception and design, Acquisition of data, Drafting or revising the article; DJW, Acquisition of data, Analysis and interpretation of data, Drafting or revising the article; CWR, EC, AJT, Acquisition of data, Analysis and interpretation of data; JSB, CJLM, Acquisition of data, Drafting or revising the article; KK, TJ, TWS, Analysis and interpretation of data, Drafting or revising the article; FM, Acquisition of data, Contributed unpublished essential data or reagents

### Author ORCIDs

Jane P Messina, http://orcid.org/0000-0001-7829-1272
Moritz UG Kraemer, http://orcid.org/0000-0001-8838-7147
Simon I Hay, http://orcid.org/0000-0002-0611-7272

# Additional files

## Major datasets

The following datasets were generated:

| Author(s) | Year | Dataset title | Dataset URL | Database, license, and accessibility information |
|---|---|---|---|---|
| Jane P Messina, Freya M Shearer | 2016 | Global compendium of human Zika virus occurrence | http://dx.doi.org/10.6084/m9.figshare.2573629 | Publicly available at figshare |
| Jane P Messina, Moritz UG Kraemer, Oliver J Brady, David M Pigott, Freya M Shearer, Daniel J Weiss, Nick Golding, Corrine W Ruktanonchai, Emily Cohn, John S Brownstein, Kamran Khan, Andrew J Tatem, Thomas Jaenisch, Thomas W Scott, Simon I Hay | 2016 | Environmental suitability for Zika virus transmission | http://dx.doi.org/10.6084/m9.figshare.2574298 | Publicly available at figshare |

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
