## [Decision Letter]

Thank you for submitting your article "Mapping global environmental suitability for Zika virus" for consideration by *eLife*. Your article has been reviewed by three peer reviewers, one of whom, Mark Jit, is a member of our Board of Reviewing Editors, and another is Jean-Paul Chretien. The evaluation has been overseen by Prabhat Jha as the Senior Editor.

The reviewers have discussed the reviews with one another and the Reviewing Editor has drafted this decision to help you prepare a revised submission.

The reviewers all felt that this was a timely analysis that provides important first estimates that are needed in light of the ongoing spread of Zika virus into new areas via infected travelers. They also commended the manuscript for increasing the epidemiological relevance by translating global temperatures into model-based estimates of suitability for dengue transmission as a surrogate for Zika virus. These are imperfect and based on static, climatic summaries of conditions that vary seasonally, but are a reasonable choice given the gaps in our present knowledge base. The reviewers also appreciated the open sharing of the datasets used.

However, the reviewers had two major concerns about the manuscript:

a) The definition of limits where "either *Ae. aegypti* probability of occurrence is more than 0.8 or temperature is conducive to transmission via this mosquito for at least 355 days in an average year" seems quite subjective and presumably helped to keep the mapped values from straying too far from known distributional limits. The reviewers would like you to indicate the objective criteria used to define these limits. Since any criteria are unlikely to be completely indisputable, they suggest that you consider providing some measure of uncertainty for the predicted suitability scores e.g. perhaps the distribution of suitability scores across ensembles.

b) Many (and possibly all) of the covariates in the models also apply to dengue – indeed two of them look at suitability for dengue transmission explicitly (due to lack of current understanding about physiological processes governing Zika transmission). Hence it remains unclear to the reviewers why the expected niche of Zika should be fundamentally different from that of dengue.

The reviewers did identify some differences between the Zika map and your corresponding map for dengue (Bhatt et al.), particularly in Africa. These are presumably driven by the dataset used to parameterise the model (based on Zika case reports in the literature). However, the very small (and potentially biased) body of literature on Zika virus prior to recent outbreaks may limit the accuracy of the estimates presented in the paper.

This needs to be carefully discussed: one solution would be to map the difference between your previous map for dengue (Bhatt et al.) and the map for Zika, with a discussion of the drivers for the difference and whether these seem to be justified (e.g. have at least some face validity).

In addition to the major comments, the reviewers also had some minor comments about presentational issues which we recommend you address, but which do not have to be explicitly responded to in your response letter:

a) On the whole the writing is excellent, but the Introduction is rather long. The focus should be sharpened to address the rationale for the present study. Much of the material presented is relevant, such as the connection to precedents of dengue and chikungunya, or the potential roles of various *Aedes* species in transmission, but it feels like a wandering review and the language should more directly point the reader to the relevance of each section.

b) The paragraph on cross-reactivity between dengue and Zika and newly emerging symptoms of Zika virus seems like a digression with little connection made to the broader purposes of the paper.

c) For vector competence, it would be useful to cite also very recent work emerging on *Ae. aegypti* and *Ae. albopictus*, and *Cx. quinquefasciatus*. Also to point out that all but the most recent vector competence work was done with ancestral African strains of Zika virus, not those circulating in the Americas.

d) *Ae. hensilli* should be mentioned as a critical vector on Yap.

e) When you state "wild isolations", it should be "isolations from wild mosquitoes" or similar.

f) The final statement in the Results about expected numbers of births needs additional context to understand how it was determined without the need to consult the detailed methods.

---

## [Author Response]

The reviewers all felt that this was a timely analysis that provides important first estimates that are needed in light of the ongoing spread of Zika virus into new areas via infected travelers. They also commended the manuscript for increasing the epidemiological relevance by translating global temperatures into model-based estimates of suitability for dengue transmission as a surrogate for Zika virus. These are imperfect and based on static, climatic summaries of conditions that vary seasonally, but are a reasonable choice given the gaps in our present knowledge base. The reviewers also appreciated the open sharing of the datasets used.

However, the reviewers had two major concerns about the manuscript:

a) The definition of limits where "either Ae. aegypti probability of occurrence is more than 0.8 or temperature is conducive to transmission via this mosquito for at least 355 days in an average year" seems quite subjective and presumably helped to keep the mapped values from straying too far from known distributional limits. The reviewers would like you to indicate the objective criteria used to define these limits. Since any criteria are unlikely to be completely indisputable, they suggest that you consider providing some measure of uncertainty for the predicted suitability scores e.g. perhaps the distribution of suitability scores across ensembles.

We thank the reviewers for this suggestion, and agree that our criteria should have been better explained. The final paragraph of the Results section now details the formulation of these criteria and refers to the appropriate sources. Additionally, we have added a map of uncertainty in the suitability scores to the supplementary information and made reference to it within this same paragraph.

*b) Many (and possibly all) of the covariates in the models also apply to dengue – indeed two of them look at suitability for dengue transmission explicitly (due to lack of current understanding about physiological processes governing Zika transmission). Hence it remains unclear to the reviewers why the expected niche of Zika should be fundamentally different from that of dengue. The reviewers did identify some differences between the Zika map and your corresponding map for dengue (Bhatt et al.), particularly in Africa. These are presumably driven by the dataset used to parameterise the model (based on Zika case reports in the literature). However, the very small (and potentially biased) body of literature on Zika virus prior to recent outbreaks may limit the accuracy of the estimates presented in the paper. This needs to be carefully discussed: one solution would be to map the difference between your previous map for dengue (Bhatt* et al.*) and the map for Zika, with a discussion of the drivers for the difference and whether these seem to be justified (e.g. have at least some face validity).*

We understand the reviewers’ concerns about the overlap in covariates used for modelling dengue and Zika. We have now addressed this in greater detail in the paragraph in the Discussion. While the covariates are indeed similar and would be well suited for a new iteration of the dengue map, we do not feel it is appropriate to make direct comparisons with the covariates from Bhatt et al. (2013) due to the fact that many refinements have been made to the covariates since that study. Regarding a difference map, while he Bhatt et al. (2013) dengue map is fundamentally different in that it used an evidence consensus layer to constrain predictions to areas with greater certainty of dengue transmission, the current ZIKV map shows environmental suitability for transmission, even in places where we know the disease has yet to be reported. Still, we felt it was useful to provide a difference map as the reviewers recommended, so in order to do so, we subtracted the values of the ZIKV map from those of the DENV map only in places where each had a non-zero prediction. This map is now found in the supporting information (Figure 2—figure supplement 4), and reference is made to it and the patterns shown in the paper.